Monitoring the influx of new species through citizen science: the first introduced ant in Denmark

http://orcid.org/0000-0002-1073-0221 Sheard Julie K. 1 julie.sheard@sund.ku.dk
Sanders Nathan J. 2
Gundlach Carsten 3
Schär Sämi 4
Larsen Rasmus Stenbak 5
1 Center for Macroecology, Evolution and Climate, Globe Institute, University of Copenhagen , Copenhagen , Denmark
2 Environmental Program, Rubenstein School of Environment and Natural Resources, University of Vermont , Burlington, VT , USA
3 Department of Physics, Technical University of Denmark , Copenhagen , Denmark
4 Unaffiliated , Dietikon , Switzerland
5 Section for Ecology and Evolution, Department of Biology, University of Copenhagen , Copenhagen , Denmark
Huber Dezene
Electronic publication date: 2020 Apr 8
Publication date: 2020
Volume: 8
Electronic Location ID: e8850
Received 2019 Oct 21; Accepted 2020 Mar 3
Copyright: © 2020 Sheard et al.
Copyright year: 2020
Copyright holder: Sheard et al.
License: This is an open access article distributed under the terms of the Creative Commons Attribution License, which permits unrestricted use, distribution, reproduction and adaptation in any medium and for any purpose provided that it is properly attributed. For attribution, the original author(s), title, publication source (PeerJ) and either DOI or URL of the article must be cited.
License URL: https://creativecommons.org/licenses/by/4.0/

Keywords: Exotics, Invasives, Citizen science, Tetramorium immigrans, Niche modelling, Species monitoring, Formicidae, DNA analysis, MicroCT, Tetramorium caespitum

Funding: Danish National Research Foundation DNRF96 15th of June Foundation, Knud Højgaard Foundation, Augustinus Foundation and Beckett Foundation Technical University of Denmark This work was supported by the Danish National Research Foundation (DNRF96), 15th of June Foundation, Knud Højgaard Foundation, Augustinus Foundation and Beckett Foundation and the 3D Imaging Centre at the Technical University of Denmark. The funders had no role in study design, data collection and analysis, decision to publish, or preparation of the manuscript.

==============================
Climate change and invasive species threaten biodiversity, yet rigorous monitoring of their impact can be costly. Citizen science is increasingly used as a tool for monitoring exotic species, because citizens are geographically and temporally dispersed, whereas scientists tend to cluster in museums and at universities. Here we report on the establishment of the first exotic ant taxon (Tetramorium immigrans) in Denmark, which was discovered by children participating in The Ant Hunt. The Ant Hunt is a citizen science project for children that we ran in 2017 and 2018, with a pilot study in 2015. T. immigrans was discovered in the Botanical Garden of the Natural History Museum of Denmark in 2015 and confirmed as established in 2018. This finding extends the northern range boundary of T. immigrans by almost 460 km. Using climatic niche modelling, we compared the climatic niche of T. immigrans in Europe with that of T. caespitum based on confirmed observations from 2006 to 2019. T. immigrans and T. caespitum had a 13% niche overlap, with T. immigrans showing stronger occurrence in warmer and drier areas compared to T. caespitum. Mapping the environmental niches onto geographic space identified several, currently uninhabited, areas as climatically suitable for the establishment of T. immigrans. Tetramorium immigrans was sampled almost three times as often in areas with artificial surfaces compared to T. caespitum, suggesting that T. immigrans may not be native to all of Europe and is being accidentally introduced by humans. Overall, citizen scientists collected data on ants closer to cities and harbours than scientists did and had a stronger bias towards areas of human disturbance. This increased sampling effort in areas of likely introduction of exotic species naturally increases the likelihood of discovering species sooner, making citizen science an excellent tool for exotic species monitoring, as long as trained scientists are involved in the identification process.

Introduction

The introduction and establishment of new species outside of their native range, which then go on to become invasive, threaten biodiversity (IPBES, 2019). Monitoring introduced and invasive species (species that have been introduced to an area outside of their range by humans and as invasives have a detrimental impact on nature, economy or human health) has been challenging in general, but has become increasingly problematic and important with accelerating trade, commerce (Meurisse et al., 2019) and climate change (Bellard et al., 2013). Recently, citizen science is being called on as a potential tool for successfully monitoring biodiversity on a large scale (Tulloch et al., 2013; Theobald et al., 2015; Sauer et al., 2017). Citizen science can help to document introduced species in general, but particularly in some of the habitats typically missed by traditional surveys.

Cities and harbours are both hotspots of introduction. In cities, conditions are typically warmer due to the urban heat island effect (Oke, 1973) making cities ideal starting points for the spread of new species (Von der Lippe & Kowarik, 2007). Likewise, harbours are a major introduction pathway (O’Connor & Weston, 2010), especially of ants (Suarez, McGlynn & Tsutsui, 2010). Yet studies of ant communities in urban ecosystems are rare (but see Lessard & Buddle, 2005; Toennisson, Sanders & Klingeman, 2011). In Denmark, 80% of foreign trade goes through Danish industrial harbours (Danske Havne, 2017). Among other products, these ships carry agricultural products, fodder and fertilisers through which new species can be introduced. While general surveys in cities and harbours can be difficult, these are some of the habitats most available to citizen scientists.

Social insects, including ants, are among the most damaging invaders (Holway et al., 2002). Over 150 ant species have been introduced outside their native range and five are listed on the world’s 100 most invasive species list by the Invasive Species Specialist Group (ISSG) (Lowe et al., 2000; Sanders & Suarez, 2011). Invasions by non-native ants have been shown to be destructive to ecosystems and financially costly to humans (Holway et al., 2002; Del Toro, Ribbons & Pelini, 2012). Although Denmark currently has no established and widespread invasive ants, thirty non-native species have so far been recorded (Schär, Illum & Larsen, 2017), most of them in hothouses in cities.

In 2017 and 2018, with a pilot study in 2015, children, in schools or with their families, collected ants for a project called the Ant Hunt (In Danish, ‘Myrejagten’) at 792 sites in Denmark. Although not the main aim of the project, we predicted that due to the expected amount of sampling in urban areas, new species for Denmark were likely to be discovered.

Here we report on the discovery of a newly established ant taxon for Denmark through the citizen science project ‘the Ant Hunt,’ Tetramorium immigrans (Santschi, 1927). T. immigrans has recently been raised to the level of a cryptic sister species of the morphologically similar T. caespitum (Wagner et al., 2017). An even more recent study has demonstrated that T. immigrans and T. caespitum hybridize (Cordonnier et al., 2019), making identification difficult.

Tetramorium caespitum and T. immigrans are both palearctic species (Wagner et al., 2017). However, while T. caespitum is common across all of Europe (Seifert, 2007), T. immigrans is primarily found in the Mediterranean, Western Europe, Central Europe, the Balkans, Eastern Europe, Anatolia and Caucasus (Wagner et al., 2017), is more thermophilic and has a more southern and urban distribution than T. caespitum (Seifert, 2018; Wagner et al., 2017). Using climatic niche modelling, we compare the climatic niche and habitat use in Europe for these two taxa, in order to determine potential differences in their ecological preferences. Finally, we compare the distance to cities and harbours along with sampled habitat types for data collected through the Ant Hunt with data collected by scientists from 1990 to 2015 to determine the extent to which data collection by citizens is or is not poised to help document introductions and shifting distributions.

Materials and Methods

Biological data

During the Ant Hunt, families and schools across Denmark collected ants by conducting baiting experiments at a site of their choosing. Participants ranged across all ages, but the average participants were children aged 5–11 years accompanied by a grown-up aged 31–50. They set out six different resources (saltwater, sugar water, oil, dissolved protein powder, a cookie and water) on bait cards and waited for two hours before collecting all the ants that were foraging on the cards. Ants were then frozen and counted before they were placed in 96% ethanol and sent to the Natural History Museum of Denmark for identification. All experiments were registered in an online database with date and GPS-coordinates (see Supplemental Material S1 for detailed protocol). In total, families and schools completed 792 experiments, of which 566 contained ants.

The ants were identified using a variety of taxonomic keys (Collingwood, 1979; Seifert, 2007; Douwes et al., 2012; Lebas et al., 2016; Wagner et al., 2017). In total, participants had collected 16,985 specimens from 29 species (Table 1). Of these, specimens from two experiments could not be identified to species level due to missing body parts and specimens from two experiments were flagged as potentially new taxa for Denmark. These were T. immigrans and Technomyrmex albipes. The establishment of Technomyrmex albipes could not be confirmed. However, after the original discovery of T. immigrans during the Ant Hunt, trained scientists resurveyed the location several times throughout 2015–2019. The presences of T. immigrans was confirmed during every survey and the species was seen to expand to an area of approx. 40 m, with more than 20 nest entrances along the pavement.

Table 1 List of the number of experiments each species was found in during the pilot study of the Ant Hunt in 2015 and the full study, which ran throughout April–September in 2017 and 2018.

Species	Experiments	
Formica cinerea	11	
Formica exsecta	1	
Formica fusca	89	
Formica lugubris	1	
Formica picea	1	
Formica polyctena	43	
Formica pratensis	1	
Formica pressilabris	1	
Formica rufa	7	
Formica rufibarbis	6	
Formica sanguinea	2	
Formica sp.2	1	
Formica truncorum	2	
Hypoponera punctatissima	1	
Lasius flavus	23	
Lasius fuliginosus	12	
Lasius niger	354	
Lasius platythorax	47	
Lasius psammophilus	2	
Lasius umbratus	1	
Myrmicinae sp.2	1	
Myrmica lobicornis	2	
Myrmica rubra	37	
Myrmica ruginodis	35	
Myrmica rugulosa	6	
Myrmica sabuleti	4	
Myrmica scabrinodis	4	
Myrmica schencki	3	
Technomyrmex albipes1	1	
Tetramorium caespitum	26	
Tetramorium immigrans1	1	
Notes:

1 New taxa for Denmark.

2 Individuals that were not determined to species level.

An experiment consisted of a baiting trial, where salt, sugar, olive oil, protein powder, water and cookies were left out for two hours after which all ants that had recruited to the baits were collected.

Tetramorium immigrans (Figs. 1A and 1B) tends to have a larger overall body size, denser striation/sculpturation of the head, thorax and petiolar nodes, as well as a more pronounced microscopic scale pattern on the first gastral tergite than T. caespitum (Wagner et al., 2017). Because of the difficulty in distinguishing T. immigrans from other species in the T. caespitum complex, we visually inspected all specimens found of T. caespitum in the Ant Hunt and randomly selected 10 samples from a broad range of localities (Fig. 1C). These were then compared to the 67 specimens of Tetramorium workers from the Botanical garden in Copenhagen, which is part of the Natural History Museum of Denmark (Fig. 1D), which were examined visually for morphological characters distinguishing typical forms of T. immigrans and T. caespitum. Voucher specimens were stored at the Natural History Museum of Denmark (NHMD 0000188537).

Figure 1 Tetramorium immigrans in Denmark.

(A) Photo of T. immigrans specimen from the Botanical Garden of Copenhagen. Taken by Rasmus S. Larsen and edited to remove background. Scale: 1 mm. (B) CT-scan of T. immigrans by Carsten Gundlach, 3D Imaging Center, DTU. Scale 0.7 mm. A video of the scan is available as Video S1. (C) Map of Denmark showing analysed samples of T. caespitum (filled orange crosses, 10 localities), observed localities of T. caespitum (open orange crosses, 83 localities), the location of T. immigrans (blue star) and localities of Ant Hunt experiments where neither T. caespitum nor T. immigrans was found (open red circles, 735 experiments). (D) Zoom in of the Botanical Garden at the Natural History Museum of Denmark in Copenhagen from Google Maps. Red circles indicate locations of T. immigrans.

One specimen was chosen to be inspected by X-ray micro computed tomography (MicroCT). The specimen was placed on a designed holder and placed inside a Zeiss Xradia 410 versa system. The system was operated with a high voltage of 40 kV and a power of 10 W. The data acquisition consisted of 3,201 images while rotating 360 degrees, each image had a pixel size of 4.14 µm. All data was reconstructed into a 4 mm by 4 mm by 4 mm 3D volume with a voxel size of 4.14 µm. The reconstructed image is shown in Fig. 1B and raw data is available through Figshare (Gundlach et al., 2020).

DNA analysis

For the DNA analysis of T. immigrans we selected two specimens from the confirmed find in the Botanical Gardens of Copenhagen, one from 2015 and one from 2018 and further selected 14 specimens of presumed T. caespitum, of which 10 had known coordinates and the remaining four were known to be from somewhere in Denmark.

Up until DNA extraction, all samples from the Ant Hunt were kept in 96% ethanol in a freezer, while samples from the Natural History Museum of Denmark had been kept as pinned specimens. We extracted DNA by cutting off a small piece of the middle leg of each specimen, to which we added 100 µl of 10% Chelex in Tris-HCI buffer. This was mixed and centrifuged for 10 min, after which the solution was heated to 99 °C for 15 min and centrifuged again. The supernatant was used as a template for PCR reactions. We used primers LCO1490 and HCO2198 (Folmer et al., 1994) to amplify mitochondrial COI gene and primer D2B and D3A-r (Saux, Fisher & Spicer, 2004) to amplify nuclear 28S rDNA gene. PCR reactions were carried out using RedTaq ReadyMix PCR Reaction Mix with 100 µg/mL Bovine serum albumin. The PCR reaction conditions consisted of an initial denaturing step of 94 °C for 5 min, followed by 35 cycles of 94 °C for 40 s, 48 °C (LCO1490/HCO2198) or 56 °C (D2B/D3A-r) for 40 s, and 72 °C for 60 s, and finally an extension step at 72 °C for 5 min. PCR products were purified using Invitek PCR clean-up MSB spin PCRapace kit and Sanger sequenced in both directions using the Mix2Seq from Eurofins.

Molecular identification was carried out by comparing samples to reference sequences from Wagner et al. (2017) (COI) and Schär et al. (2018) (28S). Raw sequences were edited and aligned using the software geneious v. 2019.0.4. A maximum likelihood tree with the best-fit model automatically selected by modelfinder and 1,000 rapid bootstrap replications was created for the alignment of COI sequences, using the program ‘IQ-TREE’ v. 1.6.1 (Nguyen et al., 2015).

Tetramorium immigrans has previously been found to be distinguished from similar species (including T. caespitum) by having a one base insertion (C) at site 438 of the 28S rDNA fragment amplified by the primers D2B and D3A-r (Saux, Fisher & Spicer, 2004; Schär et al., 2018). We aligned the sequence of the specimen from Copenhagen to the reference sequences mentioned above to see if the characteristic insertion of T. immigrans is present.

All genetic data referred to in this publication are available via the European Nucleotide Archive project PRJEB36036.

Climatic niches of T. immigrans and T. caespitum

We compared the environmental niche and predicted geographical suitability based on climate of T. immigrans and T. caespitum in Europe using occurrence data from AntMaps (Guénard et al., 2017) and the Ant Hunt along with 10 climatic variables from Worldclim 1.4 at 2.5 arc-min (~5 km) resolution (Hijmans et al., 2005).

Due to the recent distinction between T. caespitum and T. immigrans (Schlick-Steiner et al., 2006; Wagner et al., 2017), we only used data points from 2006 onwards from trusted AntMaps sources (Borowiec & Salata, 2018b; Espadaler, Pradera & Santana, 2018; Schär et al., 2018; Wagner et al., 2017, 2018). This resulted in 739 samples of T. caespitum and 187 samples of T. immigrans. Since T. immigrans was detected in only one location in Denmark, this one location does not contribute much to the overall niche analysis for the species.

We selected the climatic variables mean annual temperature, annual temperature seasonality, mean temperature of warmest quarter, mean temperature of coldest quarter, mean annual precipitation, annual precipitation seasonality, precipitation of wettest quarter, precipitation of driest quarter, precipitation of warmest quarter and precipitation of coldest quarter because these have been suggested to be the most relevant to T. immigrans (Schlick-Steiner et al., 2006; Steiner et al., 2008). We then tested for autocorrelation between the 10 climatic variables through a correlation matrix (Fig. S1) using the R packages ‘Hmisc’ (Harrell & Dupont, 2019) and ‘PerformanceAnalytics’ (Peterson & Carl, 2019). There was a strong correlation between mean annual temperature and mean temperature of warmest quarter (r = 0.90, p < 0.001) and mean temperature of coldest quarter (r = 0.94, p < 0.001). Annual precipitation was strongly correlated with precipitation of wettest quarter (r = 0.95) and precipitation of driest quarter (r = 0.86, p < 0.001). Precipitation of warmest quarter and precipitation of coldest quarter were also strongly correlated (r = 1, p < 0.001). We therefore discarded mean temperature of warmest quarter, mean temperature of coldest quarter, precipitation of wettest quarter, precipitation of driest quarter and precipitation of coldest quarter from the analysis and kept the remaining five variables.

Using the ecospat package (Broennimann, Cola & Guisan, 2018) in R Core Team (2018), we compared the environmental niches of T. caespitum and T. immigrans in a gridded environmental space, where each cell corresponds to a unique set of the five climatic variables. We first calculated the density of occurrences and the climatic variables along two climatic axes of a multivariate analysis and then measured the niche overlap along the gradients of this multivariate analysis as in Broennimann et al. (2012). Niche overlap was calculated using Schoener’s overlap metric ‘D’ (Schoener, 1968, 1970), which in this case compares the frequency of observations for each species within the chosen climatic categories (Schoener, 1968). However, we acknowledge that there are other ways to measure niche overlap (Seifert, 2017).

To test for niche equivalency and niche similarity, we compared the observed D metric with 1,000 simulated values of D. The niche equivalency test determines whether the overlap between the niches of the two species is higher than two random niches drawn from the same data pool. The niche similarity test determines whether the overlap between the two niches is higher than when one species’ niche is randomly drawn in the study area (Broennimann et al., 2012). If the observed value of D falls within the density of 95% of the simulated values, there is no detectable difference in the climatic niche of T. immigrans and T. caespitum. Finally, we projected the environmental niche of T. immigrans and T. caespitum onto Europe to visually compared the two species and pinpoint areas of suitable climate for T. immigrans to establish.

Habitat differences

We used the CORINE 100 × 100 m land cover raster dataset (Copernicus, 2012) and extracted land cover values for all data points for both species using the spatial analysis tool ‘extract values to points’ in ArcGIS (ESRI, 2010). The CORINE land cover dataset consists of 48 land cover types. For this analysis we excluded all data points that were labelled with no data or one of the water based land cover types (‘water bodies,’ ‘water courses,’ ‘sea and ocean’). The remaining 39 land cover types were reduced to six major classes (‘Artificial surface,’ ‘Agriculture,’ ‘Forest,’ ‘Scrub and/or herbaceous vegetation associations,’ ‘Open spaces with little or no vegetation’ and ‘Wetlands’) following the CORINE land cover nomenclature (Copernicus, 2015).

To test whether sampling of T. immigrans and T. caespitum were spatially biased, we compared the fraction of samples within each of the six land cover classes with the fraction of these land cover types in Europe using chi-square tests. We then did the same comparing the two species to each other to determine if T. immigrans and T. caespitum were being sampled in different habitats.

Monitoring for new species

In order to determine the suitability of citizen science for early detection and monitoring of introduced species, we compared the citizen scientist collected dataset of the Ant Hunt with a dataset collated from the Natural History Museum of Denmark, a personal collection by Sämi Schär and the Ph.D. course EuroAnts, which was collected in Denmark from 1990 to 2015, with 2015 being the most recent year with available data. We compared the datasets based on two measures, (1) distance to likely introduction sites and (2) sampling effort in different land cover types. We calculated the average distance from data points in each dataset to Denmark’s seven major cities (cities of 50.000 + inhabitants) and 31 major harbours (harbours with a yearly goods turnover of one million ton) in ArcGIS (ESRI, 2010) and compared the two datasets using a Mann Whitney U-test.

For the land cover analysis, we used the above-mentioned major categories, but also included the category Water as an approximation of samples collected close to the shore of water bodies (Copernicus, 2015). We summarised the number of samples collected in these seven major land cover classes and compared the observed values with expected values based on the availability of each land cover class using Chi-square tests. Based on availability of each land cover class we calculated the ratio of observed samples to expected samples to determine how much more or less a specific land cover type was sampled than what would be expected by chance.

Results

Tetramorium immigrans

Tetramorium immigrans was first discovered in the Botanical Garden of the Natural History Museum of Denmark (55.69, 12.57) during a pilot run of the citizen science project in 2015 and was confirmed to be established and spreading in 2018 and 2019.

Although we did not collect traditional morphometric data, we regard the find of T. immigrans in Denmark to be verified. We base this determination on a maximum likelihood tree of the mitochondrial COI gene (Fig. 2A), the existence of a characteristic one base insertion (C) in the 28S rDNA fragment (Fig. 2B), and visual examination of diagnostic characters. This extends the northern limit of T. immigrans in Europe by three degrees latitude from Gmina Janowiec Wielkopolski, Poland (52.73, 17.50, Borowiec & Salata, 2018a) to Copenhagen, Denmark (55.69, 12.57), almost 460 km.

Figure 2 Molecular identification of T. immigrans from the Botanical Garden of Copenhagen.

(A) A maximum likelihood tree for the mitochondrial COI gene. The reference alignment is the full alignment from Wagner et al. (2017) (757 sequences available from GenBank). The red arrow shows the position of the sequence from Copenhagen within mtDNA lineage E, predominantly consisting of T. immigrans. Other clades within the tree have been collapsed for simplicity. (B) 28S rDNA alignment (T. immigrans) showing a characteristic insertion of (C) at position 438 of the fragment amplified by the primers D2B and D3A-r, apparently not shared by T. caespitum and other members of the T. caespitum complex (Schär et al., 2018). The lowest sequence and chromatogram are from the sample from Copenhagen, showing the typical insertion (C) of T. immigrans at site 438.

Climatic niches of T. immigrans and T. caespitum

Environmental niche overlap between T. immigrans and T. caespitum, measured as D, which compares the frequency of observations for each species within the chosen climatic categories, was 13% with only a slight difference of the niche centroid in environmental space (Fig. 3A). Based on the contribution of the five climatic variables along the two axes, T. caespitum is present in colder and wetter areas, compared to T. immigrans, which prefers warmer and drier conditions with stable temperatures (Fig. 3B). T. immigrans and T. caespitum differed in mean climatic values for eight of the original ten climatic variables (Table S1; Fig. S2), with no difference in precipitation seasonality and precipitation of coldest quarter. Despite these slight differences, the two species’ niches were more similar to each other than two randomly drawn niches within the same data pool (niche equivalency test, p = 1) and more similar than when a random niche of either species was drawn within the available climatic space (niche similarity test, p = 0.22).

Figure 3 Climatic niche of Tetramorium caespitum and Tetramorium immigrans.

(A) Climatic niche of T. caespitum (orange) and T. immigrans (blue) along the two first axes of the PCA analysis. Light blue indicates niche space occupied by both species. Black and dashed contour lines indicate 100% and 75% available European (background) climate. The arrow marks the difference between the niche centres of the two species. (B) Correlation circle indicating the contribution of the five climatic predictors to the PCA axes of A. (C) Geographical model of climatic suitability of Europe for T. caespitum. Black dots are data points used in the analysis. (D) Geographical model of climatic suitability of Europe for T. immigrans. Black dots are data points used in the analysis.

Projection of the climatic niche into geographical space shows Eastern and Central Europe to be the most climatically suitable for both species along with the northern part of Southern Europe and the southern part of Northern Europe (Figs. 3C and 3D), although T. immigrans may have a slightly more southern distribution than T. caespitum.

Habitat differences

The current available data does not allow for an exact determination of the land use of T. caespitum and T. immigrans, but we can determine in which land cover types the two species are currently observed. Tetramorium caespitum was mostly observed in forest and agricultural land cover types (34.1% and 34.96% of observations, respectively, Table 2). T. immigrans was mostly found in land types with artificial surfaces (48.89%, Table 2). Both species were rarely found in wetland areas (T. immigrans: 1.11% and T. caespitum 0.14% of observations). Observations of both species differed significantly from what would be expected if land use reflected availability of the six land cover types in Europe (Chi square tests, χ2 = 322.71, df = 5, p < 0.001 for T. immigrans and χ2 = 243.95, df = 5, p < 0.001 for T. caespitum).

Table 2 Overview of the distribution of observation points for T. caespitum and T. immigrans across six broad land cover categories.

Land cover categories were derived from CORINE 2012 and their availability was summarized across Europe. The ratio of occurrences in each land cover type by the two species is calculated as T. caespitum: T. immigrans. Occurrence data was received from antmaps.org and combined with the samples from the Ant Hunt. A value higher than 1 indicates that the land cover type was used proportionally more by T. caespitum and a value lower than 1 indicates that the land cover type was used proportionally more by T. immigrans.

Land cover class	Europe	T. caespitum	T. immigrans	Species ratio	
%	Counts	%	Counts	%	
Artificial surfaces	4.14	121	17.34	88	48.89	0.35	
Agriculture	43.03	244	34.96	47	26.11	1.34	
Forest	30.06	238	34.1	24	13.33	2.56	
Scrub/herbaceous vegetation	14.12	84	12.03	16	8.89	1.35	
Open with little-no vegetation	6.06	10	1.43	3	1.67	0.88	
Wetlands	2.61	1	0.14	2	1.11	0.13	
Total	100	698	100	180	100		

The two species also showed significant difference in the number of observations in each land cover type compared to each other (Chi-square test, χ2 = 82.66, df = 5, p < 0.001). T. immigrans was found almost three times more often in land cover types with artificial surfaces than T. caespitum (ratio: 0.35; Table 2). On the other hand, T. caespitum was sampled over twice as often in forests than T. immigrans (ratio 2.56; Table 2).

Monitoring for new species

Overall, data collected by citizen scientists during the Ant Hunt was significantly closer to cities than data collected by scientists from 1990 to 2019 (mean 31.27 km ± 29.72 SD and 34.52 km ± 23.21 SD, respectively, W = 127,623, p < 0.001). For the Ant Hunt, 101 of 667 observations (15%) were within Denmark’s major cities compared to only 4 out of 448 (0.9%) scientist-collected samples. Data from the Ant Hunt was also significantly closer to major harbours than data from scientists (20.66 km ± 14.90 SD and 28.25 km ± 12.62 SD, respectively, W = 98,154, p < 0.001). Only six of the major harbours in Denmark were outside of cities with more than 5,000 residents (Fig. 4), suggesting a high correlation between industrial harbours and residential areas.

Figure 4 Distribution of observations by scientists and citizen scientists in Denmark.

Samples of ants in Denmark collected by scientists are in purple and samples collected by citizen scientists through the Ant Hunt are in yellow. The location of the major harbours are marked by blue triangles and cities with over 5,000 residents are marked as red polygons. Six of the major cities used in this study are labelled by name. The seventh, Frederiksberg, is not, because it is a city within København (Copenhagen).

Both scientists and citizen scientists were significantly biased regarding which land cover classes they sampled within (Chi-square test, χ2 = 653.75, df = 6, p < 0.001 and χ2 = 3,094.4, df = 6, p < 0.001, respectively). However, although both datasets were biased towards areas with artificial surfaces, the effect was far more pronounced among citizen scientists, who sampled artificially surfaced areas eight times more than expected. Scientists only sampled artificial surface areas three times more than expected by random sampling. Both citizens and scientists avoided agricultural areas, but scientists sampled forests, scrub, coastal areas and wetlands 2–3 times more than expected by random sampling (Table 3).

Table 3 Sampling effort across land cover types.

Observed (Obs.) number of samples collected within each land cover type by scientists and citizen scientists (CS) and the expected (Exp.) value if samples had been collected in accordance to availability. The ratio column refers to the ratio of observed: expected number of samples, a value of 1 would mean that a land cover type is sampled equally to its availability.

Land cover class	Denmark	Scientists (n = 448)	CS (n = 667)	
Proportion	Obs.	Exp.	Ratio	Obs.	Exp.	Ratio	
Agriculture	0.76	112	338	0.33	145	504	0.29	
Artificial surface	0.08	107	34	3.14	429	51	8.46	
Forest	0.09	139	39	3.53	54	59	0.92	
Open with little-no veg.	0.00	0	0	0.00	1	1	1.50	
Scrub/herbaceous veg.	0.04	53	17	3.11	30	25	1.18	
Waterside	0.02	21	10	2.13	5	15	0.34	
Wetlands	0.02	16	9	1.79	3	13	0.22	

Discussion

Genetic comparison of the Danish T. immigrans samples with samples from Wagner et al. (2017) (COI) and Schär et al. (2018) (28S) confirmed that T. immigrans is established in Denmark. However, genetic analysis of 14 additional Tetramorium samples from Denmark collected during the Ant Hunt in 2017 and 2018 conclude that, so far, the distribution of T. immigrans in Denmark is limited to the Botanical Garden of the Natural History Museum of Denmark.

The climatic niche of the two species overlapped by 13% and our study confirms previous assessments that T. immigrans prefers warmer and drier climates than T. caespitum (Seifert, 2018). Mean temperature of warmest quarter (°C) was 20.27 ± 2.06 SD for T. immigrans and 16.77 ± 2.43 SD for T. caespitum. This is in accordance with previously recorded standard air temperatures (°C) for May–August for both species (19.9 ± 2.5 and 16.1 ± 2.0 SD, respectively; Wagner et al., 2017).

There was a large discrepancy between identified climatically suitable areas based on the climatic niche model and current known distribution of T. immigrans. This discrepancy is in accordance with previous models of the climatic niche for T. immigrans (Steiner et al., 2008). Coupled with the knowledge that northern observations of T. immigrans have been largely within cities and the large gap in the known distribution from Poland to Denmark, we deem it likely that T. immigrans is not native to northern Europe and is being accidentally introduced by humans.

Others have also hypothesized that T. immigrans may not be native in most of Europe. Specifically, the species is thought to be introduced in France, Germany and Poland (Gippet et al., 2017; Borowiec & Salata, 2018a; Seifert, 2018; Cordonnier et al., 2019). If true, its’ current observed distributional focus in Southern Europe, along with the highest COI variability being in Anatolia and the Caucausus region (Wagner et al., 2017) could be an indication of its’ origin.

Whether T. immigrans will be able to further establish and spread in Central, Northeastern and Northern Europe will depend on a number of factors common for establishment success, including propagule pressure (Lockwood, Cassey & Blackburn, 2005) and competition with pre-established species (Menke et al., 2007). It may well be, that T. immigrans will not be able to spread beyond cities in its’ northern range, which is a common pattern for introduced species (King & Porter, 2007). Certainly, places identified as climatically suitable by the environmental niche model should be monitored closely (Fig. 3D) and further studies on the species’ distribution, competitive ability and climatic tolerances would be of high value to determine the risk of spread.

While not a main goal of the Ant Hunt, the finding of this species shows how the engagement of untrained volunteers, even children, can be a great asset to the monitoring of biodiversity, especially when it comes to detecting newly introduced species. This is evident from the sample bias of citizen scientists towards cities, harbours and areas of high human disturbance. On the other hand, scientists are more prone to sampling in natural areas. We argue the case that engaging the aid of amateur participants; even as young as 5–11 year-old children, can be a valuable tool for biodiversity monitoring. Citizen scientists are best able to search for species where scientists are most likely to miss them and where introductions are most likely.

Conclusions

We hypothesized that, although not necessarily the main goal of citizen science projects, these projects have a high likelihood of turning up new species, due to the large amount of sampling being carried out in areas of likely introduction, such as harbours and cities.

During the Ant Hunt, a citizen science project, where children set out baiting experiments to help understand the community composition and resource requirements of ants under different environmental conditions, two new species were discovered. One species, T. immigrans, was determined to be established in the Botanical Garden of the Natural History Museum of Denmark in Copenhagen.

Our findings push the distribution of T. immigrans north by almost 460 km. Our subsequent analysis of the climatic niche and potential geographical distribution of T. immigrans adds some support to the current speculation that this species may not be native to all of Europe and is spreading through introduction to cities, with many currently uninhabited locations identified as climatically suitable. A systematic survey of the land cover preferences of T. immigrans along with a genetic mapping is needed to fully map and understand the dispersal of T. immigrans across Europe.

Supplemental Information

Supplemental Information 1 Mean values for each species across the ten climatic variables and W and p values from unpaired two-sample Wilcoxon test.

Click here for additional data file.

Supplemental Information 2 Correlation matrix of the ten climatic variables initially considered for use in the climatic niche model.

Click here for additional data file.

Supplemental Information 3 Density plots of the ten climatic variables for Tetramorium caespitum (orange) and Tetramorium immigrans (blue).

Click here for additional data file.

Supplemental Information 4 Description and protocol for the citizen science project ‘the Ant Hunt’.

Click here for additional data file.

Supplemental Information 5 A 319 bp alignment of COI gene sequences overlapping with the DNA-barcoding region from 757 specimens of Tetramorium from Wagner et al. (2017), and one specimen of T. immigrans from Copenhagen (last sequence).

Click here for additional data file.

Supplemental Information 6 Raw GPS data for T. immigrans and T. caespitum from Antmaps and the Ant Hunt combined.

Data points from Denmark were collected by the authors. Data points from other countries were downloaded from https://antmaps.org/ on 08-08-2019.

Click here for additional data file.

Supplemental Information 7 A 638 bp alignment of 28S rDNA sequences of 6 specimens belonging to 5 species of the Tetramorium caespitum complex from Schär et al. (2018) and one specimen of T. immigrans from Copenhagen (last sequence).

Click here for additional data file.

Supplemental Information 8 Climatic suitability based on a reduced T. caespitum dataset.

Illustrates that removing data from potentially questionable sources does not affect the conclusion of the model.

Click here for additional data file.

We would like to thank all the children and citizen scientists who helped collect the data for the Ant Hunt. A special thanks to the children at Sølvgade School, who first discovered T. immigrans in Denmark.

Additional Information and Declarations

Competing Interests

Author Contributions

DNA Deposition

Data Availability

The authors declare that they have no competing interests.

Julie K. Sheard conceived and designed the experiments, performed the experiments, analyzed the data, prepared figures and/or tables, authored or reviewed drafts of the paper, and approved the final draft.

Nathan J. Sanders conceived and designed the experiments, performed the experiments, authored or reviewed drafts of the paper, and approved the final draft.

Carsten Gundlach conceived and designed the experiments, performed the experiments, prepared figures and/or tables, authored or reviewed drafts of the paper, and approved the final draft.

Sämi Schär conceived and designed the experiments, performed the experiments, analyzed the data, prepared figures and/or tables, authored or reviewed drafts of the paper, and approved the final draft.

Rasmus Stenbak Larsen conceived and designed the experiments, performed the experiments, analyzed the data, prepared figures and/or tables, authored or reviewed drafts of the paper, and approved the final draft.

The following information was supplied regarding the deposition of DNA sequences:

The sequence is available at GenBank: PRJEB36036.

The following information was supplied regarding data availability:

Data for T. immigrans and T. caespitum in Europe are available as a Supplemental File, but we encourage researchers interested in using the data to contact https://antmaps.org/ as this is a dataset that is continuously updated.

Data from the Ant Hunt citizen science project in Denmark is available at: Koch Sheard J. (2020). The Danish Ant Hunt. Center for Macroecology, Evolution and Climate, University of Copenhagen. Sampling event dataset DOI 10.15468/dcijnc accessed via GBIF.org on 2020-03-07.

Genetic data is available at GenBank: PRJEB36036.

Supplemental video material is available at: Gundlach, Carsten; Sheard, Julie K. (2020): Tetramorium immigrans visualisation files. figshare. Media. DOI 10.11583/DTU.10000637

Code can be found on GitHub: https://github.com/JKSheard/tetramorium-DK

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
