# Peer review of "Monitoring the influx of new species through citizen science: the first introduced ant in Denmark"

_PeerJ, doi:10.7717/peerj.8850_

## Round 0.1 · original submission · Minor Revisions

First of, thank you to the three reviewers (two of whom provided public reviews) for their work on this MS. I really appreciate the time that you took to apply your expertise to improving this paper.

Second, thanks to the authors for submitting an interesting and relevant manuscript. With the increase in use of citizen science methods, and the impact of biological invasions, this paper is bound to be well-cited.

I am applying a decision of "minor revisions" to this version because I think that the authors can accomplish the suggested changes within that general context. The two biggest issues that I can see are outlined by Reviewer #2 – the possibility of hybrids (point 3, and partially point 2), and queries about the climate niche model (point 8).

While this is a minor revision recommendation, because those two issues are particularly important for this manuscript, I may decide to send it out for one more round of review following revisions.

·

Basic reporting

The manuscript is well written and other than a very few instances where sentence structure seems a bit ambiguous (and they are noted in the specific comments), the English is excellent.

The manuscript examines:
-the use of citizen collectors to more broadly sample the environment for ants and compares those efforts to those of biologists.
-the appearance of a new species (T. immigrans) to Denmark and the role of citizen science in making that discovery
-the habitat and climatic niche differences between two species of Tetramorium in Europe.

The literature is well and appropriately referenced.

The structure is professional.

The Figures and raw data seem appropriate. I note in general comments that the quality of Figure 3 b was not great in my copy. I also note that I am not sure how Figure 1 b contributes to the manuscript. It is a fascinating and technically interesting image, but does not seem to advance the manuscript intent.

As a general comment, I did find (and this could arise from my initial quick reading) that the distinction between Danish (essentially presence/absence data for T. immigrans as it was only in one location) and the broader European dataset was not clear, at least for me. It wasn’t until I read that T. immigrans was found in only one location in Denmark that Denmark wasn’t really contributing to the overall niche analysis for the species. If this single discovery was noted earlier and clearly, it would be easier to understand the limited role of the Denmark T. immigrans data in the European analysis.

Experimental design

The manuscript meets my reading of the ‘Aims and Scope’ of Peer J.

The research questions are clear (although note my last comment under Basic Reporting).

The technical aspects are excellent.

With respect to the ability to replicate the research, I would ask for more details about the protocol given to students for collections. It might be also noted how the schools were approached to participate.

Validity of the findings

The results arise from a well designed methods. I would only mention the need to mention earlier the single finding of T. immigrans in Denmark.

Conclusions appropriate to the results

Additional comments

Some editorial comments

Line 23: Do you mean, “Here we report on the discovery of Tetramorium immigrans, the first established exotic ant taxon in Denmark through The Ant Hunt,” or ‘Here we report on the discovery of Tetramorium immigrans, the first exotic ant taxon in Denmark established through The Ant Hunt.’

Line 65: What about the Hypoponera?

Line 86 (Biological data).
-It would be useful to know the age range of students participating
-It would be useful to know what six foods were used on the bait cards (I see this in Table 1 but this is likely to be placed in ‘Results’ which is distant from where the protocol in introduced)
-Is this project only ant focused or part of a larger citizen science project
-Is this protocol part of an established monitoring protocol. If so, what protocol (reference?)

Line 110 (Paragraph): I am not sure of the purpose of this imaging as only one specimen was used. Imaging of multiple specimens of the two cryptic species would seem of use for taxonomists, but I am uncertain if the one image is of value and should be included.

Line 122: Did you have students directly place specimens into 96% ethanol?

Line 147: you don’t explicitly mention the use of PCA in these methods but provide that in Figure 3

Line 154: some statement regarding confidence in correct IDs might help here given the noted difficulties in distinguishing between the two species.

Line 170-175: I will just note that while many editors do not like explanations of methodological analyses, many readers find them quite useful.

Para 240 and 245: First paragraph (Line 240) notes you can distinguish differences in land cover types which suggests those cover types are about to be noted (but do not seem to be). The next paragraph (Line 245), states “The two species also ….” which suggests you are moving onto different aspects of land cover. Am I missing something?

Line 238: “…more than expected.” More than expected from what? Random sampling?

Table 1 title: Would the term ‘sampling block’ be better than ‘experiment?’ I suggest ‘block’ given that a variety of foods were used. The term ‘sampling block’ may not be quite right, but experiment suggests a protocol with a control.

Table 1: It would be good to indicate which other species are known exotics (e.g., Hypoponera).
Figure 2: “b) 28S rDNA alignment showing a characteristic insertion of T. immigrans …”
Might be better as
“b) 28S rDNA alignment (T.immigrans) showing a characteristic insertion of (C) …”

Figure 3: the quality of Figure 3 b is not very good in the copy I have received.

Figure 3: the meaning of the shading in c and d are not noted in the title

·

Basic reporting

I put all my comments for the editor and the authors under "General comments for the author".

Just briefly:
English is great.
article structure, figures, tables are fine (same small mistakes like missing spaces in legends).
some literature missing (details see below).
Raw data: morphometric data of Tetramorium immigrans workers from Denmark should be shared!

Experimental design

I put all my comments for the editor and the authors under "General comments for the author".

Just briefly:
Original primary research within Aims and Scope of the journal: yes.
Research question well defined, relevant & meaningful: yes.
Methods well described.

Validity of the findings

I put all my comments for the editor and the authors under "General comments for the author".

Just briefly:
The huge difference of the climate model and the real distribution of Tetramorium immigrans is a problem. Maybe the model should be made new to be meaningful (details below).

The study has the potential to show one important novelty better: an overview where in Europe Tetramorium immigrans might be native and where it is introduced.

Additional comments

Dear Julie K. Sheard et al.,

It is very interesting to investigate the invasive potential of Tetramorium immigrans. There are several other authors who already speculated about that aspect (as you have correctly cited in the manuscript).

However, I think there are some aspects in the manuscript which should be improved.

1) You have written: “… may not be native to all of Europe and is being accidentally introduced by humans“
In this aspect I see the chance to show something very interesting new. It would be great to give an overview where in Europe it might be native and where it is introduced. For the situation in Austria, for example, T. immigrans occurs in natural and seminatural habitats around Vienna (or in the east and south of Vienna), but not in natural habitats in Styria (around Graz), where it is only semisynanthropic (WAGNER 2011). The situation is similar around Lyon (France) or Poland, where it might be a non-native (GIPPET & al. 2017, BOROWIEC & SALATA 2018, CORDONNIER & al. 2019). I would guess that T. immigrans is native at least for the Balkans and the Pannonia. It is a mentioned to be a Neozoon for Germany (SEIFERT 2018). Maybe based on published habitat information (e.g., WAGNER & al. 2017) you can work out some information about its native range. You should mention somewhere that the highest COI variability is in Anatolia and Caucausus region (WAGNER & al. 2017). Thus, most likely there is its origin.

2) You have written: “However, an even more recent study demonstrated that T. immigrans and T. caespitum widely share the same gene pool when occurring in sympatry (Cordonnier et al., 2019), challenging this idea.”
Hybridization does not challenging species delimitation between T. immigrans and T. caespitum! They are good species. Please take a look into papers of modern species concepts (DE QUEIROZ 2007, SEIFERT 2014) to consider that hybridization is allowed between species and Mayr’s old concept (MAYR 1942) is just an idea far away from species reality.

3) It is often underestimated how difficult the identification of Tetramorium is. The lack of morphometric data of the T. immigrans nest in Denmark is a weakness and with your data we cannot fully exclude the possibility of a hybrid. The combination of COI and 28S is enough to separate pure caespitum from pure immigrans. But the question regarding a putative hybrid nest is still open and thus an alternative hypothesis to the occurrence of immigrans. Interspecific hybridization rates are higher in anthropogenic influenced habitats (SEIFERT 1999), where T. immigrans in Western/Central(/Northern) Europe often occurs. Especially in an area so far in north, where immigrans might be very rare (in relation to caespitum), the force do hybridize interspecifically might be high for an immigrans gyne. What I see on https://vimeo.com/user86430849 shows me that with POTCos of around 11 (only unilateral) probability of dealing with T. immigrans and not with T. caespitum is higher (but I cannot fully exclude hybrids). At the moment nobody is fully able to identify such hybrids (the Kaufmann group did that genetically). Seifert did that a few times morphometrically or tried to do that, but he mentioned that hybrids are similar to T. staerckei (which would not consider a problem in this study, since nobody would suggest that non-invasive T. staerckei occurs in Denmark). You should have used the key of WAGNER et al. (2017) and/or SEIFERT (2018) in addition to genetics, which demands morphometric data. Put all these morphometric data into the publication. If you are dealing with a hybrid, maybe future scientists will be able to proof that by your published morphometric data.

4) “Niche overlap was calculated using Schoener’s overlap metric…“
Whenever you write about niche overlap in ants you have to cite Seifert’s monograph of ant ecology, which also deals with the calculation of ecological niches in ants (SEIFERT 2017).

5) You should mention that the TAS (mean temperature from May to August) was before published for all 10 European species of the caespitum complex (WAGNER & al. 2017, Tab. 4) and the values for immigrans and caespitum are very similar with your “Mean temp. of warmest quarter”.

6) The fact that the ant material was collected via baiting by schools and not via hand-collecting from nests by myrmecologists is not perfect for the scientific aspect of your paper. You could have individuals from different nests in the same sample. Usually you should work with nest samples in Tetramorium.

7) “… making citizen science an excellent tool for exotic species monitoring”
I do not agree with that. Citizens might be useful to detect birds etc., but myrmecology needs accuracy. I rather consider it as problematic that citizens will, for example, confuse samples (which is also a big problem for myrmecologists). So I suggest toning down this message and tell the reader about possible risks. By the way: concerning several details about citizen scientists at all: I don’t think that these things are worth to be published that detailed in PeerJ (but that should be decided by the editor and not by me).

8) I am not expert for climate niche modeling; however, the suggested niche of T. immigrans in Fig. 3 d) does not convince me. According this niche, eastern Germany must be THE hot spot of T. immigrans. However, not a single record is known from this well-investigated area. This “hot spot” might be an artifact based on the use of urban records of T. immigrans. In cities the temperature is higher than in the surroundings. Shouldn’t such records be omitted to model a climate niche to mean anything? On the other hand, Mediterranean areas in, for example, Greece or Croatia (were T. immigrans is very common at the coast) are white in your map, which does not reflecting the reality.
In that content, you only have written: “It has already been hypothesized that T. immigrans may not be native in most of Europe (Gippet et al., 2017; Borowiec & Salata, 2018; Cordonnier et al., 2019). If true, its’ current observed distributional focus in Southern Europe could reflect invasion history rather than climatic preference. This would explain why T. immigrans is presently less widespread in northern Europe than environmental niche modelling would predict (Fig. 3d).”
I think this is not enough to discuss to huge difference between the model and the real distribution.

And a few small things:

1) Line 99: “T. albipes“: use full genus name.

2) “T. caespitum/impurum complex“ is the old name for the complex, you should use the new name “T. caespitum complex”.

3) Fig. 2a: “lineage E“ not in italics. And what does “~” mean?

4) Fig. 2b is very interesting!!

5) Go through the whole manuscript for detecting sloppinesses like missing spaces, missing full stops etc. Here an example: Table 1: You must never write “sp.“ in italics.

6) Legend of Tab. 2: Tell the reader also in the legend from which source you have taken this data.

I hope you consider my critics as constructive and that you will improve the manuscript.

Literature:
BOROWIEC, L. & SALATA, S. 2018: Tetramorium immigrans Santschi, 1927 (Hymenoptera: Formicidae) nowy gatunek potencjalnie inwazyjnej mrówki w Polsce. – Acta entomologica silesiana 26: 1–5.
CORDONNIER, M., GAYET, T., ESCARGUEL, G. & KAUFMANN, B. 2019: From hybridization to introgression between two closely related sympatric ant species. – Journal of Zoological Systematics and Evolutionary Research.
DE QUEIROZ, K. 2007: Species concepts and species delimitation. – Systematic Biology 56: 879-886.
GIPPET, J.M.W., MONDY, N., DIALLO-DUDEK, J., BELLEC, A., DUMET, A., MISTLER, L. & KAUFMANN, B. 2017: I’m not like everybody else: urbanization factors shaping spatial distribution of native and invasive ants are species-specific. – Urban Ecosystems 20: 157-169.
MAYR, E. 1942: Systematics and the origin of species from the viewpoint of a zoologist. – Harvard University Press, Cambridge, 334 pp.
SEIFERT, B. 2014: A pragmatic species concept applicable to all eukaryotic organisms independent from their mode of reproduction or evolutionary history. – Soil Organisms 86: 85-93.
SEIFERT, B. 1999: Interspecific hybridisations in natural populations of ants by example of a regional fauna (Hymenoptera, Formicidae). – Insectes Sociaux 46: 45-52.
SEIFERT, B. 2018: The ants of Central and North Europe. – lutra Verlags- und Vertriebsgesellschaft, Tauer, 408 pp.
SEIFERT, B. 2017: The ecology of Central European non-arboreal ants – 37 years of a broad-spectrum analysis under permanent taxonomic control. – Soil Organisms 89: 1-67.
WAGNER, H.C. 2011: Tag der Artenvielfalt – Die Ameisen (Hymenoptera: Formicidae) im Botanischen Garten Graz. – Mitteilungen des naturwissenschaftlichen Vereines für Steiermark 141: 235–240.
WAGNER, H.C., ARTHOFER, W., SEIFERT, B., MUSTER, C., STEINER, F.M. & SCHLICK-STEINER, B.C. 2017: Light at the end of the tunnel: integrative taxonomy delimits cryptic species in the Tetramorium caespitum complex (Hymenoptera: Formicidae). – Myrmecological News 25: 95-130.

Reviewer 3 ·

Basic reporting

Presented research is supported by solid analyses, and I strongly recommend its publication. The manuscript is written in clear and professional English. However, the introduction is, in some places, inconsistent and misleading. Its greater part deals with introduced and invasive species. But the Authors didn't point out differences between those two groups. It is also not clear to which of those group Tetramorium immigrans is assigned, and why? In further parts of the ms, the Authors started also using the term “exotic.” Is it possible to explain the relationship between those three terms? In my opinion, “exotic” usually refers to species originated from the biogeographic realm different from the one in which it was collected. All evidence suggests that Tetramorium immigrans is a Palearctic species, thus describing it as “exotic” can be confusing.
Additionally, information referring to the species is, in my opinion, very scarce, and it lacks background data of distribution and biology of the whole caespitum group (Wagner et al., 2017 published comprehensive revision of the entire group). The Authors stated that Tetramorium immigrans “is thought to have a more southern and urban distribution than T. caespitum,” but they failed to describe the distribution range of T. caespitum. I think that, in this case, distribution ranges of both taxa should be discussed in detail to show discussed differences.
Lines 100-103 very clearly point out differences between T. immigrans and T. caespitum. However, the mentioned characters are not visible on photographs 1a and 1b. Especially photograph 1a is blurry and indistinct. To visualize discussed differences, the Authors should also include photographs of T. caespitum.

Experimental design

In lines 148-149, the Authors stated that the geographical distribution of T. immigrans and T. caespitum in Europe was implemented from AntMaps. In my opinion, this approach is quite risky. On the cited website is stated that “There are errors and uncertainties that are inherited from the data sources and others that might have been introduced during the databasing process. Distributional data in GABI should be viewed critically before forming the basis of scientific investigation.” Did the Authors check if used records have been published or verified by experts? If so, they should cite the original sources of those records.
Line 308: the statement that the “species may not be native to all of Europe” needs clarification. What does it mean? If not to all, then which part can be considered as native, and why?

Validity of the findings

no comment

---

## Round 0.2 · Minor Revisions

Thank you to the referee who completed a second review of this paper. The application of his expertise will doubtless improve this paper.

There are still a few minor issues – in particular with the climate niche model and distribution (southern Spain and southern Greece). Please respond and/or revise. I will conduct the final editorial review and consideration of the response for your upcoming revision.

·

Basic reporting

The English is better than mine.

Literature references are sufficient.

Ad raw data: It should be written in the manuscript that traditional morphometric data of the T. immigrans workers in Denmark have NOT been collected. Based on that, every reader can decide by himself if he or she trusts in the identification. By the way, all in all, for me the identification is convincing. (I do not know if anybody is able to transfer a CT-scan video into absolute traditional morphometric values. So that video probably does not help a lot.)

Ad figures: I have to complain that I am still not very happy with the climate niche model of T. immigrans (Fig. 3d), although the authors have mentioned the aspects of city data (which is welcome!). I have mentioned the problem with eastern Germany in the first review just as one unlucky example. But the model is also inappropriate for other parts of Europe. What is about England, for example? Phil Attewell is interested in finding new English Tetramorium species for years, but he has not found T. immigrans. According this model (Fig. 3d), eastern England shoud be and ideal area for the species. But, empirically, that is not the case, probably because of climatic reasons, temperature and/or precipitation. I have no good idea how to improve the problematic aspect that the model does not fit with the realitiy. The authors should think once again if they made any mistake in the data analyses and how they can explain such unexpected results. Is the city aspect enough?

Experimental design

Original primary research: Yes.

Research question well defined, relevant & meaningful: Yes.

Rigorous investigation performed to a high technical & ethical standard: Unfortunately, I consider one big problem which I have not recognized during the first review (and which has been already considered by Rev. 3): The Tetramorium caespitum data of antmaps.org are partly not convincing and probably in parts wrong. I do not believe that Tetramorium caespitum occurs on southern Greek islands as well as in some low-altitude parts of southern Spain near the coast. This records are most likely identification errors and the data should not be used for the analyses. Who did the determination of these samples? Usually, Tetramorium caespitum is replaced by other species in the very south of Europe at low altitudes. The authors should think more critical about the published data they use and omit those which are not convincing and maybe do the analyses new.

Validity of the findings

All fine (but see 1. and 2.).

Additional comments

Dear Julie K. Sheard et al.,

Thank you for your detailed answers to all my comments! I consider that you thought about them all. I like your answers and many of them convince me.

I am sorry that I still see some problems in the current version of the manuscript and I hope you can rather solve the problems or convince me or the editor that the problems do not exist.

Here some small aspects:

Line 59: One „.“ is missing.

Line 83: Shouldn’t there be a comma after „Anatolia“? As I have heard a comma before „and“ shoud always be there in a list of three or more words.

Line 168: Put „immigrans“ in lower case.

Line 308: I do not understand the followig sentence: „This would also explain the discrepancies between its’ current distribution and modelled distribution based on its climatic niche, which suggests the species should be present in eastern Germany (Fig. 3d).“ I do not understand the explanation, what is „This“?

If you are interested in my record of Tetramorium immigrans in the Botanical Garden of Graz, look here:
https://www.zobodat.at/pdf/MittNatVerSt_141_0235-0240.pdf
I called the species "hemisynanthropic".

Take care and best wishes,
Heri

---

## Round 0.3 · accepted · Accept

Thank you to the reviewers and the co-authors for making this a smooth process. I believe that the thoughtful reviews have served to improve this interesting manuscript, and I look forward to seeing it in press. Congratulations.

I would like to encourage the authors to publish the review history alongside the manuscript, as I believe that this will add substantial value to this work for researchers and students alike. But, of course, that is up to the co-authors.